# Cross-sectional study evaluating data quality of the National Cancer Registration and Analysis Service (NCRAS) prostate cancer registry data using the Cluster randomised trial of PSA testing for Prostate cancer (CAP)

Samuel William David Merriel,[1] Emma L Turner,[1] Eleanor Walsh,[1] Grace J Young,[1] Chris Metcalfe,[1] Luke Hounsome,[2] Isobel Tudge,[2] Jenny Donovan,[1] Freddie Hamdy,[3] David Neal,[3] Richard M Martin[1]

[1]Population Health Sciences, Bristol Medical School, University of Bristol, Bristol, UK
[2]Public Health England National Cancer Registration and Analysis Service, Bristol, UK
[3]Nuffield Department of Surgical Sciences, Oxford University, Oxford, UK

**Correspondence to**
Dr Samuel William David Merriel;
sam.merriel@bristol.ac.uk

## ABSTRACT

**Objectives** To compare the completeness and agreement of prostate cancer data recorded by the National Cancer Registration and Analysis Service (NCRAS) with research-level data specifically abstracted from medical records from the Cluster randomised triAl of prostate specific antigen (PSA) testing for Prostate cancer (CAP) trial.

**Design** Cross-sectional comparison study.

**Participants** We included 1356 men from the CAP trial cohort who were linked to the NCRAS registry.

**Primary and secondary outcome measures** Completeness of prostate cancer data in NCRAS and CAP and agreement for tumour, node, metastases (TNM) stage (T1/T2; T3; T4/N1/M1) and Gleason grade (4–6; 7; 8–10), measured by differences in proportions and Cohen's kappa statistic. Data were also stratified by year and pre-2010 versus post-2010, when NCRAS reporting standards changed.

**Results** Compared with CAP, completeness was lower in NCRAS for Gleason grade (41.2% vs 76.7%, difference 35.5, 95% CI 32.1 to 39.0) and TNM stage (29.9% vs 67.6%, difference 37.6, 95% CI 34.1 to 41.1). NCRAS completeness for Gleason grade (pre-2010 vs post-2010 31.69% vs 64%; difference 32.31, 95% CI 26.76 to 37.87) and TNM stage (19.31% vs 55.50%; difference 36.19, 95% CI 30.72 to 41.67) improved over time. Agreement for Gleason grade was high (Cohen's kappa, κ=0.90, 95% CI 0.88 to 0.93), but lower for TNM stage (κ=0.41, 95% CI 0.37 to 0.51) overall. There was a trend towards improved agreement on Gleason grade, but not TNM stage, when comparing pre-2010 and post-2010 data.

**Conclusion** NCRAS case identification was very high; however, data on prostate cancer grade was less complete than CAP, and agreement for TNM stage was modest. Although the completeness of NCRAS data has improved since 2010, the higher completeness rate in CAP demonstrates that gains could potentially be achieved in routine registry data. This study's findings highlight a need for improved recording of stage and grade data in the source medical records.

### Strengths and limitations of this study

► Cross-sectional study design.
► Comparison of English prostate cancer registry with independently collected, research-level data from a randomised controlled trial on prostate cancer screening.
► Low level of completeness of tumour, node, metastases staging data in National Cancer Registration and Analysis Service (NCRAS) overall.
► Reporting standards for NCRAS registry changed in 2010, affecting the consistency of data collection and quality across the study period.

## INTRODUCTION

Accurate and complete cancer registry data can inform health policy, prioritise the allocation of limited funds available for cancer prevention and treatment, and be used for research purposes. Bray and Parkin[1 2] outline methods for evaluating the quality of cancer registry data, including comparisons with other local cancer registries, routinely collected data, large-scale research cohorts or estimates using incidence rates.

Prostate cancer is the most common cancer in males in England, with 39 741 new cases registered in 2014.[3] The National Cancer Registration and Analysis Service (NCRAS) collects data on new cases of cancer in England,[4] including prostate cancer. The NCRAS was established in February 2016 to incorporate the National Cancer Registry Service and the National Cancer Intelligence Network, and is now part of Public Health England. The 'Cancer reform strategy'[5] and the 'Improving outcomes: a strategy

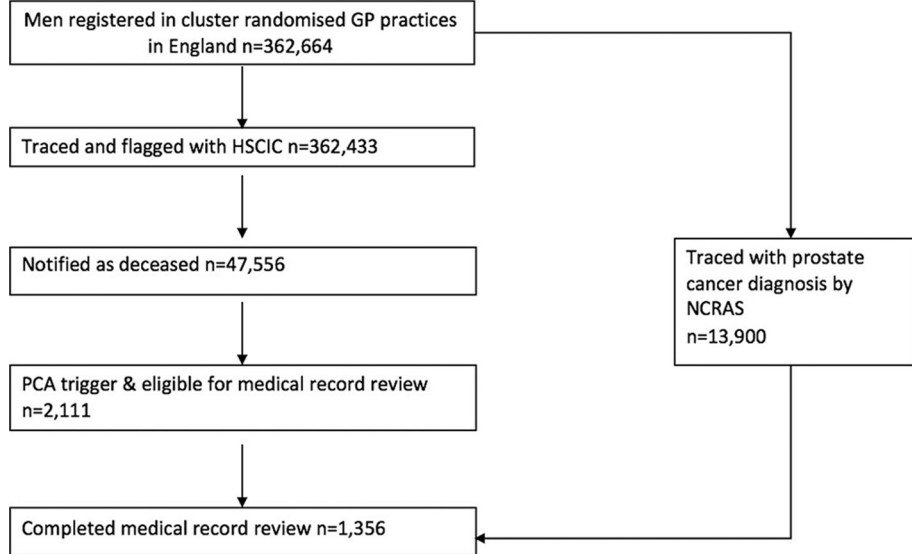

**Figure 1** Participant recruitment flow diagram. HSCIC, Health and Social Care Information Centre; NCRAS, National Cancer Registration and Analysis Service; PCA, Prostate Cancer.

for cancer'[6] reports from the Department of Health in England highlighted the need for better information and higher quality data to support efforts to improve cancer outcomes. The National Cancer Dataset was subsequently replaced by the Cancer Outcomes and Services Dataset to improve cancer registry data-collection standards in England.[7] There is some grey literature on the quality of the NCRAS cancer registry data, showing low levels of completeness of tumour, node, metastases (TNM) stage.[8] However, to our knowledge, comparison studies against independently collected data have not yet been published for prostate cancer.

The Cluster randomised triAl of PSA testing for Prostate cancer (CAP) trial is a population-based cluster randomised controlled trial in England and Wales assessing the impact of a single invitation for PSA testing for prostate cancer on prostate cancer mortality.[9] Within this trial, trained researchers extract detailed data from the medical records of men who have died with or of prostate cancer onto a structured pro forma. These data offered the opportunity to compare the completeness and agreement of the NCRAS cancer registry data with an independent research-level data source.

## MATERIALS AND METHODS
### Participant selection
NCRAS collate patient-identifiable demographic and cancer-specific clinical information based on the Clinical Outcomes and Services Dataset from National Health Service (NHS) service providers, unless the patient opts out.

The CAP trial involves 416 387 men aged 50–69 years from 271 primary care general practitioner practises in England and Wales that were randomised between 2001 and 2007.[9] In this trial, these men were flagged with the Health and Social Care Information Centre (HSCIC) to trace cancer and death registrations. The comparison with the NCRAS data restricts the analysis to those from CAP practices in England who were deceased. In England, 362 433 CAP men were successfully traced by the HSCIC and 47 556 were notified as having died of any cause. Of the men successfully traced and who had died, 2111 were identified from HSCIC data as having a prostate cancer diagnosis or having prostate cancer listed anywhere on the death certificate. These men underwent medical record review by specially trained CAP research staff to ascertain cause of death independent from information held on the death certificate and the cancer registry.[9] At the time of this analysis, 1356 medical record reviews had been completed for men diagnosed with prostate cancer between February 2002 and December 2015, who had died between April 2003 and October 2015. These men were matched to their NCRAS dataset entry using their NHS number (see figure 1), then given a unique identifier and merged.

### Data
The date of birth provided by the general practitioner practicses at the time of recruitment was used to calculate age at entry into the CAP study and age at diagnosis.

The NCRAS team extracted available diagnostic data, including date of diagnosis, American Joint Committee on Cancer[10] and Union Internationale Contre le Cancer (UICC) TNM staging,[11] and Gleason grade[12] for the current analysis. Clinical TNM stage was derived from available imaging data, and pathological TNM stage was derived from pathology reports of tissue samples if surgery was within 6 months of diagnosis, with final TNM staging altered if necessary. NCRAS Clinical TNM stage data were combined with pathological stage data by the NCRAS team where it was incomplete to produce a 'best' stage, and these data were used in the analysis.

The data collected for CAP from the medical records included date of prostate cancer diagnosis, diagnostic Gleason grade and diagnostic clinical UICC TNM stage. These data were extracted for the current analysis, in addition to digital rectal examination findings, radiology reports and PSA blood results at the time of diagnosis. The results of these investigations were used in the current analysis to derive clinical TNM stage if this was missing from the data extraction. Diagnostic clinical stage, grade and other investigations were selected by taking the closest record to the date of diagnosis within ±6 months.

TNM stage data gathered for CAP did not always yield a complete TxNxMx stage from the clinical record. A pragmatic, cascading hierarchical approach was therefore used if this situation occurred. Any evidence of advanced disease (T4 or N1 or M1) was gathered initially, followed by evidence of locally advanced disease (T3) and then localised disease (T1 or T2). This was done separately for NCRAS and CAP stage data.

This study follows the Strengthening the reporting of observational studies in epidemiology (STROBE) reporting guidelines for observational studies (STROBE checklist submitted as online supplementary file 1).

### Statistical analysis

All analyses were performed using STATA V.14 (StataCorp, 2015). We compared the completeness of Gleason grade and TNM stage between NCRAS and CAP data using differences in proportions (with 95% CIs). The agreement between the NCRAS data and CAP data was assessed using Cohen's kappa statistic, with 95% CIs, for Gleason grading and TNM staging. We also undertook an analysis comparing localised (T1/2) versus locally advanced (T3) versus advanced (T4 or N1 or M1) prostate cancer. Reporting standards for new cases to the registry changed in 2010–2011,[7] so the completeness of data was also stratified by year of prostate cancer diagnosis and 2002–2009 vs 2010–2015.

### RESULTS

The average age at the time of diagnosis of both the NCRAS and CAP participants was 75.2 years (SD 5.1).

97.9% (n=1327) of men with prostate cancer notified to the NCRAS registry had a complete diagnosis date. Gleason grade was entered for 41.2% (95% CI 38.6% to 43.8%) of NCRAS men. UICC TNM stage was entered for 29.9% (95% CI 27.5% to 32.5%) (table 1).

Of the CAP trial men with complete medical record reviews, 98.5% (n=1336) had complete date of birth and diagnosis date. The overall completeness of Gleason grade data and TNM stage was 76.7% (95% CI 74.4% to 78.9%) and 67.6% (95% CI 65.0% to 70.0%), respectively.

The date of diagnosis in the NCRAS registry exactly matched CAP trial men in 44.5% (n=583) of cases. The median difference in diagnosis date for NCRAS patients was 0 day (IQR 0–7) later than CAP men. NCRAS had significantly lower overall completeness than CAP for

**Table 1** Completeness of Gleason grade and TNM stage in NCRAS registry and CAP

| n=1356 | Combined Gleason grade | | | | TNM stage | | | |
|---|---|---|---|---|---|---|---|---|
| Year(s) | NCRAS (n, %) | CAP (n, %) | Difference (%) | 95% CI | NCRAS (n, %) | CAP (n, %) | Difference (%) | 95% CI |
| 2002–2003 | 22  26.19 | 74  88.10 | −61.91 | −73.58 to 50.23 | 7  8.33 | 56  66.67 | −58.33 | −70.02 to 46.65 |
| 2004 | 25  33.33 | 59  78.67 | −45.34 | −59.47 to 31.20 | 18  24.00 | 50  66.67 | −42.67 | −57.06 to 28.27 |
| 2005 | 41  33.61 | 101  82.79 | −49.18 | −59.91 to 38.45 | 30  24.59 | 84  68.89 | −44.26 | −55.48 to 33.04 |
| 2006 | 47  29.94 | 129  82.17 | −52.23 | −61.57 to 42.89 | 21  13.38 | 109  69.43 | −56.05 | −65.01 to 47.09 |
| 2007 | 55  31.43 | 150  85.71 | −54.28 | −62.90 to 45.67 | 33  18.86 | 128  73.14 | −54.29 | −63.04 to 45.53 |
| 2008 | 63  34.05 | 157  84.86 | −50.81 | −59.37 to 42.25 | 32  17.30 | 114  61.62 | −44.32 | −53.20 to 35.45 |
| 2009 | 49  31.61 | 110  70.97 | −39.36 | −49.58 to 29.13 | 43  27.74 | 106  68.39 | −40.65 | −50.81 to 30.48 |
| 2010 | 93  62.84 | 105  70.95 | −8.11 | −18.79 2.57 | 53  35.81 | 89  60.16 | −24.32 | −35.36 to 13.28 |
| 2011 | 77  68.14 | 73  64.60 | 3.54 | −8.77 15.85 | 51  45.13 | 84  74.34 | −29.20 | −41.41 to 16.99 |
| 2012 | 66  64.71 | 59  57.84 | 6.87 | −6.47 20.20 | 90  88.24 | 69  67.65 | 20.59 | 9.56 to 31.61 |
| 2013–2015 | 20  54.05 | 23  62.16 | −8.11 | −30.51 14.30 | 28  75.68 | 27  72.97 | 2.70 | −17.19 to 22.60 |
| 2002–2009 | 302  31.69 | 780  81.85 | −50.16 | −53.99 to 46.32 | 184  19.31 | 647  67.89 | −48.48 | −52.47 to 44.70 |
| 2010–2015 | 256  64.00 | 260  65.00 | −1.00 | −7.63 5.63 | 222  55.50 | 269  67.25 | −11.75 | −18.45 to 5.05 |
| Overall | 558  41.15 | 1040  76.70 | −35.55 | −38.99 to 32.09 | 406  29.94 | 916  67.55 | −37.61 | −41.10 to 34.12 |

CAP, Cluster randomised triAl of PSA testing for Prostate cancer; NCRAS, National Cancer Registration and Analysis Service; TNM, tumour, node, metastases.

**Table 2** Agreement between NCRAS registry and CAP on combined Gleason grade

| n=1356 | NCRAS (n) | CAP (n) | Kappa (κ) | 95% CI |
|---|---|---|---|---|
| Combined Gleason grade (2002–2009) | | | | |
| 4–6 | 80 | 202 | 0.84 | 0.77 to 0.92 |
| 7 | 97 | 241 | 0.86 | 0.79 to 0.92 |
| 8–10 | 125 | 337 | 0.92 | 0.88 to 0.97 |
| Overall | 302 | 780 | 0.87 | 0.84 to 0.92 |
| Combined Gleason grade (2010–2015) | | | | |
| 4–6 | 37 | 28 | 0.95 | 0.89 to 1.00 |
| 7 | 90 | 93 | 0.95 | 0.90 to 0.99 |
| 8–10 | 129 | 139 | 0.97 | 0.93 to 0.99 |
| Overall | 256 | 260 | 0.93 | 0.92 to 0.94 |
| Combined Gleason grade (overall) | | | | |
| 4–6 | 117 | 230 | 0.88 | 0.83 to 0.94 |
| 7 | 187 | 334 | 0.90 | 0.86 to 0.94 |
| 8–10 | 254 | 476 | 0.94 | 0.92 to 0.97 |
| Overall | 558 | 1040 | 0.90 | 0.88 to 0.93 |

CAP, Cluster randomised triAl of PSA testing for Prostate cancer; NCRAS, National Cancer Registration and Analysis Service.

Gleason grade (difference in proportions 35.6%, 95% CI 32.1% to 39.0%), and TNM stage (37.6%, 95% CI 34.1% to 41.1%), although the difference in completeness of Gleason grade (1.00%, 95% CI −5.63% to 7.63%) and TNM stage (11.75%, 95% CI 5.05% to 18.45%) data was much smaller from 2010 onwards.

Table 2 shows agreement on Gleason grade data in the registry and the CAP trial, which was strong. Agreement was high for combined (κ=0.90, 95% CI 0.89 to 0.92) Gleason grade overall. There was a trend towards higher agreement in 2010–2015 compared with 2002–2009 for low, moderate and high combined Gleason grade. This level of agreement was maintained across individual levels of primary (κ=0.84, 95% CI 0.82 to 0.86) and secondary Gleason grades (κ=0.80, 95% CI 0.79 to 0.83).

Table 3 shows agreement on overall UICC TNM staging. The levels of agreement for overall stage accuracy were weak to moderate. Similar levels of agreement were found when considering the data categorised as localised (T1/2, κ=0.53, 95% CI 0.42 to 0.64), locally advanced (T3, κ=0.29, 95% CI 0.17 to 0.41) and advanced (T4 or N1 or M1, κ=0.47, 95% CI 0.37 to 0.57). These findings were consistent when comparing data from 2002–2009 with 2010–2015.

### DISCUSSION

There were mixed levels of completeness and agreement of stage and grade information comparing NCRAS with CAP research-level data. We showed a high level of completeness for case identification and diagnosis dates in both cohorts. NCRAS completeness for Gleason grade was moderate and TNM stage

completeness was low. From 2010, the completeness of stage and grade data in the NCRAS registry increased. Completeness in CAP remained consistently high throughout the study period. Agreement between NCRAS and CAP was high for Gleason grade and moderate for TNM stage.

The completeness of case capture and diagnosis date is consistent with the UK and Ireland Association of Cancer Registries' most recent performance indicators.[13] Our results are similar to those from validation studies of cancer registries in other countries. Estimates of completeness for registries in Ireland,[14] Sweden,[15 16] Bulgaria[17] and Norway[18] were generally high, ranging from 91% to 97%. These studies compared registry data with other population registries or primary care records. None of them used an independently collected research-level cancer trial dataset for comparison. Tomic et al[16] found a higher level of agreement in Sweden for both TNM staging (83%) and Gleason grading (97%) when compared with our study. This could be due to the lower level of grade and stage completeness in our study and changes in the UICC TNM classification during the study period, and the fact that Tomic et al compared two cancer registries that gathered prostate cancer data on the same population.

The strengths of the data used in the current analysis to compare completeness and agreement with the NCRAS cancer registry were that they were obtained through extensive medical notes review by trained researchers. Furthermore, almost every CAP trial

**Table 3** Agreement between NCRAS registry and CAP on TNM stage

| n=1356 | NCRAS (n) | CAP (n) | Kappa (κ) | 95% CI |
|---|---|---|---|---|
| TNM stage (2002–2009) | | | | |
| T1/T2 | 91 | 218 | 0.55 | 0.40 to 0.69 |
| T3 | 21 | 242 | 0.33 | 0.15 to 0.51 |
| T4/N1/M1 | 72 | 122 | 0.53 | 0.38 to 0.67 |
| Overall | 184 | 582 | 0.47 | 0.43 to 0.50 |
| TNM stage (2010–2015) | | | | |
| T1/T2 | 64 | 57 | 0.45 | 0.28 to 0.62 |
| T3 | 40 | 90 | 0.25 | 0.08 to 0.41 |
| T4/N1/M1 | 117 | 84 | 0.41 | 0.28 to 0.55 |
| Overall | 221 | 231 | 0.34 | 0.27 to 0.37 |
| TNM stage (overall) | | | | |
| T1/T2 | 155 | 275 | 0.53 | 0.42 to 0.64 |
| T3 | 61 | 332 | 0.29 | 0.17 to 0.41 |
| T4/N1/M1 | 189 | 206 | 0.47 | 0.37 to 0.57 |
| Overall | 405 | 813 | 0.41 | 0.37 to 0.51 |

CAP, Cluster randomised triAl of PSA testing for Prostate cancer; NCRAS, National Cancer Registration and Analysis Service; TNM, tumour, node, metastases.

participant's diagnosis had been notified to the NCRAS registry, allowing a thorough comparative analysis.

The completeness of UICC TNM stage data in the registry was moderate, making it more difficult to draw conclusions about the accuracy of TNM staging. A possible reason for the differences in agreement between NCRAS and CAP could be due to the fact that NCRAS more commonly partially or completely used pathological TNM staging compared with CAP. Distinguishing TNM stage of prostate cancer radiologically, particularly T1b vs T2, is an evolving area,[19] and this may explain some of the difficulties in reporting. Reporting standards for NCRAS cancer registries changed in 2010 and completeness improved after the changes were made.[7]

Our findings suggest that prostate cancer data in the NCRAS cancer registry are complete in terms of identifying and recording new cases. While the agreement for Gleason grade was high, the completeness and agreement of TNM stage data were lower for the years covered in this analysis, highlighting a need for improved recording of these data in the source medical records. This study also demonstrates how trial data can be verified for completeness and accuracy using empirical data. Completeness of NCRAS data has improved since 2010, however the higher completeness rate in CAP highlights what further gains could potentially be achieved in routine registry data. Complete and accurate national cancer registries are vital to inform health policy, healthcare spending and research.

**Acknowledgements** The authors would like to acknowledge the Public Health England National Cancer Registration and Analysis Service (South West) for their assistance with this study.

**Contributors** The Cluster randomised triAl of PSA testing for Prostate cancer (CAP) trial was performed by RMM, JD, FH, DN, ELT, CM, GJY and EW, amongst others. ELT and EW extracted the relevant CAP trial data for this study. LH and IT extracted the relevant National Cancer Registration and Analysis Service registry data for this study. SWDM, EW and GJY performed the data analysis. SWDM drafted this manuscript, to which all authors made amendments and approved the final version.

**Funding** The CAP trial was supported by Cancer Research UK and the UK Department of Health (C11043/A4286, C18281/A8145, C18281/A11326 and C18281/A15064).

**Competing interests** RM,ET, JOD, and CM have received grants from Cancer Research UK in the previous three years.

**Ethics approval** Ethical approval to access information about all those in the CAP study was obtained from UK Trent Multi-Centre Research Ethics Committee (MREC/03/4/093). The CAP study was exempt from gaining individual consent to participant having obtained Section 251 approval from the UK Patient Information Advisory Group (PIAG) (now the Confidentiality Advisory Group, CAG), under Section 251 of the NHS Act 2006 [PIAG 4-09 (k)/2003]. The CAP study is sponsored by the University of Bristol and is registered at Current Controlled Trials (ISRCTN92187251).

**Provenance and peer review** Not commissioned; externally peer reviewed.

**Data sharing statement** The dataset supporting the conclusions of this article cannot be shared with third parties due to conditions within the Section 251 approval that governs this research.

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
