## [Reviewer comments · BMJ Open]

ARTICLE DETAILS

TITLE (PROVISIONAL)	Cross-sectional study evaluating data quality of the National Cancer Registration and Analysis Service (NCRAS) prostate cancer registry data using the Cluster randomised triAl of PSA testing for Prostate cancer (CAP)
AUTHORS	Merriel, Samuel; Turner, Emma; Walsh, Eleanor; Young, Grace; Metcalfe, Chris; Hounsome, Luke; Tudge, Isobel; Donovan, Jenny; Hamdy, Freddie; Neal, David; Martin, Richard

VERSION 1 - REVIEW

REVIEWER	Dr Anna Gavin N. Ireland Cancer Registry Centre for Public Health Queens University Belfast Mulhouse Building Grosvenor Road Belfast BT12 6DP N. Ireland
REVIEW RETURNED	05-Feb-2017

GENERAL COMMENTS	This is an important area for research as the national cancer registry is used for service planning and monitoring, for evaluation of public health initiatives, for quality assessment of cancer screening programs, to provide information essential for genetic counselling services and also for the examination of possible cancer clusters. Accuracy and full data completeness of data are essential for the completion of these tasks. The method of comparison with an independently collected dataset is a recognised gold standard of data examination. The researchers treated each source in the same way looking at data items within a six month window. The researchers focus on stage and grade data items and report levels of completeness from both sources noting improvements over time in the data collected by the National Cancer Registration Service (NCRAS). This was not reflected in the title as from the title I expected more data items to be compared. The discussion transiently mentions the completeness of other data items such as date of diagnosis, however these are not documented in the results. I feel that a table to include how many patients in the CAP trial were in the NCRAS and the accuracy of data items such as date of diagnosis within 7 days, NHS number, etc would be useful if possible and would better reflect the completeness of the NCRAS in patient notification which is their primary role. The figure indicates that 2111 patients were eligible for note review however, this was completed on 1,356. The reasons for this reduction should be explained. In the discussion note should be taken that prostate cancer patients
---

	are older than the average cancer patient and that prostate cancer is not always diagnosed pathologically as some patients are kept under review while PSA levels are monitored and surgery is delayed. This may have affected the levels of completeness in the NCRAS. The tables need better labelling re the numbers. In the discussion mention is made of prostate cancer registries however as far as I am aware there are just population based cancer registries which include prostate.
--	--

REVIEWER	YANG, Won Jae Soonchunhyang University, South Korea
REVIEW RETURNED	12-Apr-2017

GENERAL COMMENTS	The reviewer completed the checklist but made no further comments.
--

REVIEWER	Markus Aly Karolinska Institutet Department of Molecular Medicine and Surgery, Sweden
REVIEW RETURNED	16-Apr-2017

GENERAL COMMENTS	Thank you for letting me review the manuscript entitled " Agreement and completeness of prostate cancer data comparing the National Cancer Registration and Analysis Service(NCRAS) and the Cluster randomised triAl of PSA testing for Prostate cancer(CAP)". It is interesting to compare research data with clinical registry data. And I think this manuscript could be used as a leverage in order to get more funding to update government funded clinical registries, which is very important in countries where a fair part of the health care is funded by taxes. General comments The manuscript is well written and has a good structure. Minor comments Introduction, page 5 row20-30 Is there any information on how complete the NCRS and the NCIN registries are? Maybe it could be exemplified by data on how complete they are. Materials and Methods Participants selection, page 6 row 14-40 It is not clear as to why the authors focus on the men who had died due to prostate cancer. In this type of study I assume the data should be easily retracted also for men alive and could therefore be compared as well. I suggest a short comment to clarify this. Data, page 7 row 6-10 It is not clear whether nor not is is the authors who combine NCRAS clinical TNM stage data with pathological stage data in order to produce `best` stage data. I am also a little bit sceptical to this, as such registry based data may be used to do prediction models later on and then the best stage data is a combination of the information you have at time of diagnosis and the information you have after
---

	treatment, which then may differ from men undergoing surgery and men undergoing radiation therapy. Results, page 8, row 10-15) The results for complete information on Gleason grade is surprisingly low in the NCRAS data. I would like a comment on that in the discussion of the manuscript – it is difficult for policymakers to draw any conclusions on trends in prostate cancer when completeness is low. Tables, page 10, to make table 1 easier to read, is it possible to greyscale every other row? Otherwise I have no comments. Discussion Adequate discussion, however I think the fact that there is fairly low coverage in both registers could be more stressed. However I think the reference used on page 14 row 5 is wrong – the article referenced does not mention this controversy, at least not to my knowledge.
--	---

REVIEWER	Moptotsugu Oya Department of Urology, Keio University School of Medicine, Tokyo Japan
REVIEW RETURNED	19-Apr-2017

GENERAL COMMENTS	Merriell S et al. in "Agreement and completeness of prostate cancer data comparing the National Cancer Registration and Analysis Service (NCRAS) and the Cluster randomized trial of PSA testing for Prostate cancer (CAP)' provide some comparison of NCRAS and CAP, but appears inadequate at the specific important observations. This report is not a study comparing the contents of the accumulated data but a report simply comparing the accumulation situation. Collecting TNM and Gleason score when conducting clinical research on prostate cancer is fundamentally important. Therefore the research itself to compare the collection situation does not make sense. For CAP trial in particular, it is necessary to reconsider the data collection method and accuracy so that it can be applied as more useful clinical data for research purpose.
--

VERSION 1 – AUTHOR RESPONSE

Reviewer(s)' Comments to Author:

Reviewer: 1

Dr Anna Gavin

N. Ireland Cancer Registry, Centre for Public Health, Queens University Belfast, N. Ireland

Please state any competing interests or state 'None declared': none declared

Please leave your comments for the authors below

This is an important area for research as the national cancer registry is used for service planning and monitoring, for evaluation of public health initiatives, for quality assessment of cancer screening programs, to provide information essential for genetic counselling services and also for the examination of possible cancer clusters. Accuracy and full data completeness of data are essential for the completion of these tasks.

The method of comparison with an independently collected dataset is a recognised gold standard of data examination. The researchers treated each source in the same way looking at data items within a six month window. The researchers focus on stage and grade data items and report levels of completeness from both sources noting improvements over time in the data collected by the National Cancer Registration Service (NCRAS). This was not reflected in the title as from the title I expected more data items to be compared.

Thank you for your kind feedback. The manuscript title has been amended to meet the journal guidelines.

The discussion transiently mentions the completeness of other data items such as date of diagnosis; however these are not documented in the results.

Please see the 4th paragraph of the Results section (Line 30-34) for data on completeness of date of diagnosis

I feel that a table to include how many patients in the CAP trial were in the NCRAS and the accuracy of data items such as date of diagnosis within 7 days, NHS number, etc would be useful if possible and would better reflect the completeness of the NCRAS in patient notification which is their primary role.

The authors did discuss whether to have a separate table for demographic data. It was decided to only include the data at the start of the Results section (see Lines 1-16).

The figure indicates that 2111 patients were eligible for note review however, this was completed on 1,356. The reasons for this reduction should be explained.

At the time this analysis was performed, there were 1,356 CAP participants who had complete medical record review data available for use in this study. We did not have enough data on the remaining 755 CAP participants. This was outlined in Lines 26-43 of the Methods section, to compliment Figure 1.

In the discussion note should be taken that prostate cancer patients are older than the average cancer patient and that prostate cancer is not always diagnosed pathologically as some patients are kept under review while PSA levels are monitored and surgery is delayed. This may have affected the levels of completeness in the NCRAS.

The tables need better labelling re the numbers.

All tables have been amended to reflect that whether the numbers are raw counts (n), proportions (%) or Kappa statistic (k)

In the discussion mention is made of prostate cancer registries however as far as I am aware there are just population based cancer registries which include prostate.

Thank you for highlighting this. The Discussion section has been adjusted accordingly.

Reviewer: 2

YANG, Won Jae

Soonchunhyang University, South Korea

Please state any competing interests or state 'None declared': None declared

Please leave your comments for the authors below

none

Thank you for reviewing this manuscript

Reviewer: 3

Markus Aly

Karolinska Institutet, Department of Molecular Medicine and Surgery, Sweden

Please state any competing interests or state 'None declared': None declared

Please leave your comments for the authors below

To the authors and editors,

Thank you for letting me review the manuscript entitled "Agreement and completeness of prostate cancer data comparing the National Cancer Registration and Analysis Service(NCRAS) and the Cluster randomised trial of PSA testing for Prostate cancer(CAP)". It is interesting to compare research data with clinical registry data. And I think this manuscript could be used as a leverage in order to get more funding to update government funded clinical registries, which is very important in countries where a fair part of the health care is funded by taxes.

General comments

The manuscript is well written and has a good structure.

Thank you for reviewing our manuscript. We appreciate your general comments.

Minor comments

Introduction, page 5 row20-30

Is there any information on how complete the NCRS and the NCIN registries are? Maybe it could be exemplified by data on how complete they are.

Comparison data from the UKIACR has been added to the second paragraph of the Discussion section.

Materials and Methods

Participants selection, page 6 row 14-40

It is not clear as to why the authors focus on the men who had died due to prostate cancer. In this type of study I assume the data should be easily retracted also for men alive and could therefore be compared as well. I suggest a short comment to clarify this.

Apologies that this was not made clear. In the CAP trial a full medical record review was only completed for deceased men. The Methods section has been amended.

Data, page 7 row 6-10

It is not clear whether or not it is the authors who combine NCRAS clinical TNM stage data with pathological stage data in order to produce 'best' stage data. I am also a little bit sceptical to this, as such registry based data may be used to do prediction models later on and then the best stage data is a combination of the information you have at time of diagnosis and the information you have after treatment, which then may differ from men undergoing surgery and men undergoing radiation therapy.

Where staging data was incomplete in the NCRAS, the NCRAS staff generated the 'best stage'. The Methods section has been amended to make this clearer.

Results, page 8, row 10-15)

The results for complete information on Gleason grade is surprisingly low in the NCRAS data. I would like a comment on that in the discussion of the manuscript – it is difficult for policymakers to draw any conclusions on trends in prostate cancer when completeness is low.

The final paragraph of the Discussion section has been amended to include this comment.

Tables, page 10,

to make table 1 easier to read, is it possible to greyscale every other row? Otherwise I have no comments.

Thank you for the suggestion. We have amended Table 1 to try and make it easier to read.

Discussion

Adequate discussion, however I think the fact that there is fairly low coverage in both registers could be more stressed.

The final paragraph of the Discussion section has been amended to include this comment.

However I think the reference used on page 14 row 5 is wrong – the article referenced does not mention this controversy, at least not to my knowledge.

This paragraph and referencing has been updated

Reviewer: 4

Moptotsugu Oya

Department of Urology, Keio University School of Medicine, Tokyo Japan

Please state any competing interests or state 'None declared': None declared

Please leave your comments for the authors below

Merriel S et al. in "Agreement and completeness of prostate cancer data comparing the National Cancer Registration and Analysis Service (NCRAS) and the Cluster randomized trial of PSA testing for Prostate cancer (CAP)' provide some comparison of NCRAS and CAP, but appears inadequate at the specific important observations.

Thank you for reviewing this article. The authors would be happy to consider any important observations that may have been missed if the reviewer would like to specify them.

This report is not a study comparing the contents of the accumulated data but a report simply comparing the accumulation situation. Collecting TNM and Gleason score when conducting clinical research on prostate cancer is fundamentally important. Therefore the research itself to compare the

collection situation does not make sense. For CAP trial in particular, it is necessary to reconsider the data collection method and accuracy so that it can be applied as more useful clinical data for research purpose.

The CAP trial was used as an independently collected, comparable dataset to assess the quality of case reporting, stage and grade data in the NCRAS cancer registry for prostate cancer. The use of independent datasets, as done in this study, is the gold standard for assessing data quality in cancer registries. The authors believe that ensuring national cancer registries have high quality data is vital to inform policy and funding decisions aimed at improving cancer outcomes.

The data collection and accuracy of the CAP trial was not the main focus of this study, and peer reviewed studies of the CAP trial will be published in due course. Further information about the CAP trial methodology can be found in the BJC paper by Turner et al (Reference 9).

VERSION 2 – REVIEW

REVIEWER	Dr Anna T Gavin Director, N. Ireland Cancer Registry Centre for Public Health Queens University Belfast Mulhouse Building Grosvenor Road Belfast BT12 6DP
REVIEW RETURNED	05-Jun-2017

GENERAL COMMENTS	In the material and methods second paragraph, second sentence it would be worth indicating at the start" In this trial these" In the conclusion the authors miss that the completeness of NCRAS is good in terms of not missing cases, it is the data items for each registered case which is not so complete, this is better summarised in the paper conclusion page 15 and the authors should consider using some of the words from that to the conclusion for the abstract
--

VERSION 2 – AUTHOR RESPONSE

Reviewer 1 comments

Thank you for reviewing our revised manuscript, and recommending it for publication

1. In the material and methods second paragraph, second sentence it would be worth indicating at the start" In this trial these"

The suggested amendment has been made to the materials and methods section

2. In the conclusion the authors miss that the completeness of NCRAS is good in terms of not missing cases, it is the data items for each registered case which is not so complete, this is better summarised in the paper conclusion page 15 and the authors should consider using some of the words from that to the conclusion for the abstract

The conclusion section of the abstract has been amended to be more consistent with the discussions and conclusions section of the main article.